# The Winning Solution to the IEEE CIG 2017 Game Data Mining Competition

**Anna Guitart** *,† [ID]**, Pei Pei Chen** †** and África Periáñez**

Yokozuna Data, a Keywords Studio, 102-0074 Tokyo, Japan; ppchen@yokozunadata.com (P.P.C.); aperianez@yokozunadata.com (Á.P.)
* Correspondence: aguitart@yokozunadata.com
† These authors contributed equally to this work.

**Abstract:** Machine learning competitions such as those organized by Kaggle or KDD represent a useful benchmark for data science research. In this work, we present our winning solution to the Game Data Mining competition hosted at the 2017 IEEE Conference on Computational Intelligence and Games (CIG 2017). The contest consisted of two tracks, and participants (more than 250, belonging to both industry and academia) were to predict which players would stop playing the game, as well as their remaining lifetime. The data were provided by a major worldwide video game company, NCSoft, and came from their successful massively multiplayer online game Blade and Soul. Here, we describe the long short-term memory approach and conditional inference survival ensemble model that made us win both tracks of the contest, as well as the validation procedure that we followed in order to prevent overfitting. In particular, choosing a survival method able to deal with censored data was crucial to accurately predict the moment in which each player would leave the game, as censoring is inherent in churn. The selected models proved to be robust against evolving conditions—since there was a change in the business model of the game (from subscription-based to free-to-play) between the two sample datasets provided—and efficient in terms of time cost. Thanks to these features and also to their ability to scale to large datasets, our models could be readily implemented in real business settings.

**Keywords:** churn; competition; video games; user behavior; behavioral data

## 1. Introduction

In video games, reducing player churn is crucial to increase player engagement and game monetization. By using sophisticated churn prediction models [1,2], developers can now predict when and where (i.e., in which part of the game) individual players are going to stop playing and take preventive actions to try to retain them. Examples of those actions include sending a particular reward to a player or improving the game following a player-focused data-driven development approach.

Survival analysis focuses on predicting when a certain event will happen, considering censored data. In this case, our event of interest is churn, and highly accurate prediction results can be obtained by combining survival models and ensemble learning techniques. On the other hand, and even though they do not take into account the censoring of churn data, deep learning methods can also be helpful to foresee when a player will quit the game. In particular, long short-term memory (LSTM) networks—recurrent neural networks (RNNs) trained using backpropagation through time—constitute an excellent approach to model sequential problems.

The 2017 IEEE Conference on Computational Intelligence and Games hosted a Game Data Mining competition [3], in which teams were to accurately predict churn for NCSoft's game Blade and Soul.

In this work, we present the feature engineering and modeling techniques that led us to win both tracks of that contest.

## 2. Game Data Mining Competition

### 2.1. The Data

The data to be analyzed came from NCSoft's Blade and Soul, a massively multiplayer online role-playing game. It consisted of one training dataset and two test datasets, for which results should be evaluated without output information. Each of the datasets covered different periods of time and had information about different users. A summary of the original records is presented in Table 1.

Thirty percent of the players churned in each of the provided data samples. A player was to be labeled as "churned" if he or she did not log in between the fourth and eighth week after the data period, as shown in Figure 1.

Log data were provided in CSV format. Each player's action log was stored in an individual file, with the user ID as the file name. In log files, every row corresponded to a different action and displayed the time stamp, action type, and detailed information. Table 2 shows a sample log file. There were 82 action types, which represented various in-game player behaviors, such as logging in, leveling up, joining a guild, or buying an item. These 82 action types were classified into 6 categories: connection, character, item, skill, quest, and guild. As shown in Table 2, the detailed information provided depended on the type of action. For example, actions related to earning or losing money included information about the amount of money owned by the character, whereas actions that had to do with battles reflected the outcomes of those battles.

**Table 1.** Details of the three datasets.

| Dataset | Data Period | Weeks | Players | Size |
|---------|-------------|-------|---------|------|
| Train | Apr-01-2017–May-11-2017 | 6 | 4000 | 48 GB |
| Test 1 | July-27-2016–Sep-21-2016 | 8 | 3000 | 30 GB |
| Test 2 | Dec-14-2017–Feb-08-2017 | 8 | 3000 | 30 GB |

**Table 2.** A sample log file.

| Time | Action Type | Details (up to 72 Columns) |
|------|-------------|----------------------------|
| 2016-04-01 00:09:30.597 | Enter world | Server ID, character ID, character job, character level, ... |
| 2016-04-01 00:12:32.158 | Join guild | Guild ID, guild level, character data, ... |
| 2016-04-01 00:14:50.084 | Enter zone | Zone ID, character data, ... |
| 2016-04-01 00:14:50.084 | Save equipped item | Item type, item level, equipment score, character data, ... |
| 2016-04-01 00:18:54.094 | Team duel end | Duel ID, duel server, duel point, type of target, ... |
| ... | ... | ... |

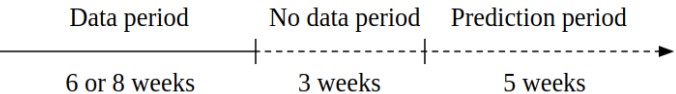

**Figure 1.** Between 6 and 8 weeks of data logs were provided (see Table 1). The goal of this competition was to predict churn and survival time from the fourth to the eighth week after the data period.

### 2.2. Challenge

The aim of this competition [3] was to predict churn and life expectancy during a certain prediction period; see Figure 1. The contest consisted of two tracks:

- In track 1, participants had to predict whether each player would log in or not during the prediction period and output binary results.

- In track 2, participants had to predict the survival time (in days) for each player, that is, the number of days between their last action in the provided data and their last predicted action.

*2.3. Evaluation*

The results of track 1 were evaluated through the $F_1$ score:

$$F_1 = 2 \times \frac{\text{precision} \times \text{recall}}{\text{precision} + \text{recall}}, \tag{1}$$

where

$$\text{precision} = \frac{\text{TP}}{\text{TP} + \text{FP}}, \tag{2}$$

$$\text{recall} = \frac{\text{TP}}{\text{TP} + \text{FN}}. \tag{3}$$

Here TP stands for true positives, FP for false positives, and FN for false negatives. On the other hand, the root mean squared logarithmic error (RMSLE) was adopted to evaluate the results of track 2:

$$\text{RMSLE} = \sqrt{\frac{1}{n} \sum_{i=1}^{n} \left( \log(o_i + 1) - \log(p_i + 1) \right)^2}, \tag{4}$$

where $o_i$ and $p_i$ are the observed and predicted survival times, respectively.

Participants should upload their prediction results for the two test sets to an online submission system, with the number of uploads limited to one every 30 min. After every submission, partial results were calculated and shown on a leader board. For these partial results, only ten percent of the players in each set were considered. The final results—calculated with the full sets—were announced after the submission system was closed.

## 3. Data Preparation and Feature Engineering

*3.1. Time-Oriented Data*

More than 3000 variables—including time series information and static inputs—were calculated for each player during the feature engineering process and incorporated into the models. Below we outline some of the computations performed.

(1) *Actions:* Number of times each type of action was performed per day. Some actions were grouped and different statistics were computed. For example, every time a player created, joined, or invited other players to a guild was counted as a "Guild" feature. The way in which actions were grouped is shown in Table 3.
(2) *Sessions:* Different sessions per player and day.
(3) *Playtime:* Total playtime per day.
(4) *Level:* Daily last level and evolution in the game.
(5) *Target level:* Daily highest battle-target level.
(6) *Number of actors:* Number of actors played per day.
(7) *In-game money:* Daily information about money earned and spent.
(8) *Equipment:* Daily equipment score. The higher this score, the better the equipment owned by the player.
(9) *Static Data:* We also calculated some statistics of the time series data (for each player), considering the whole data period.
(10) *Statistics:* Different statistics, such as the mean, median, standard deviation, or sum of the time-series features mentioned in Section 3.1, were applied depending on the variable.

For example, the total amount of money the player got during the data period or the standard deviation of the daily highest battle-target level.

*(11)*   *Actors information:* Number of actors usually played.

*(12)*   *Guilds:* Total number of guilds joined.

*(13)*   *Information on days of the week:* Distribution of actions over different days of the week.

**Table 3.** Groups of actions considered in this work.

| Group | Actions |
|---|---|
| Team | Invite a target to a team, join a team, kick out a target from a team, team battle |
| Guild | Create a guild, invite a target to a guild, join a guild |
| Quest | Complete a quest, complete daily challenge, complete weekly challenge |
| Trade (Get) | Get money and item through exchange-window |
| Trade (Give) | Give money and item through exchange-window |
| Skill | Acquire skill, level up skill, use training point, acquire quest skill |
| Item | Repair item, evolve item, exceed item limit |
| Auction | Put item to auction, buy item from auction |

### 3.2. Autoencoders

Autoencoders are unsupervised learning models based on neural networks, which are often applied for dimensionality reduction [4] and representation learning [5]. Compared to human feature selection, applying autoencoders is less labor-intensive and serves to capture the underlying distribution and factors of the inputs [6].

Autoencoders can be trained employing various kinds of neural networks and are widely used in many machine-learning areas. For example, the authors of [7] applied convolutional neural networks (CNNs) to train autoencoders for representation learning on electroencephalography recordings. A deep belief network autoencoder was used to encode speech spectrograms in [8]. LSTM autoencoders were trained to reconstruct natural language paragraphs in [9]. And another work [10] combined generative adversarial networks with autoencoders to encode visual features.

As neural-network-based encoders, the structures of autoencoders are multilayer networks with an input layer, one or several hidden layers, and an output layer. While training autoencoders, the inputs and target outputs are the same [5]. The values of the nodes in a hidden layer are the encoded features. A reconstruction error, which is measured by a cost function such as the mean squared error, is calculated to evaluate the discrepancy between the inputs and their reconstructions, that is, the values in the output layer. As the reconstruction error decreases, the ability of the encoded features to represent and reconstruct the input values increases. This minimization of the reconstruction error is usually accomplished by stochastic gradient descent [11].

## 4. Feature Selection

As commented in the previous section, there were more than 3000 extracted features after the data preparation. Besides simply adopting all these features, the following three feature selection methods were also tested.

### 4.1. LSTM Autoencoder

An LSTM autoencoder was employed to derive representative features from the time-series data described in Section 3.1. It comprised five layers: the input layer, a reshape layer, a dense (fully connected) layer, another reshape layer, and the output layer. The values of the nodes in the dense layer are the encoded features. In this work, time-series data were encoded, reducing the dimension of the feature space up to 500 features.

*4.2. Feature Value Distribution*

In the periods covered by the training and Test 1 datasets, Blade and Soul used a subscription payment system, where players pay a fixed monthly fee. In contrast, in the period corresponding to the Test 2 dataset the game was free-to-play, an approach in which players only pay to purchase in-game items. This change in the business model significantly affected player behavior: for example, most of the players in the Test 2 dataset gained much more experience than players in the training or Test 1 sets.

A feature may become noise if its distribution in the test set is too different from that in the training set. To select only those features distributed over a similar range in both the training and test sets, we calculated the 95% confidence interval for each feature in the training set. After performing experiments with cross-validation, a feature was selected only when more than 70% of its values in the test set were within that 95% confidence interval derived from the training set.

*4.3. Feature Importance*

A permutation variable importance measure (VIM) based on the area under the curve (AUC) [12] was chosen in the case of survival ensembles. This method is closely related to the commonly used error-rate-based permutation VIM and is more robust against class imbalance than existing alternatives, because the error rate of a tree is replaced by the AUC.

## 5. Machine Learning Models

In this section we describe the models that served us to win the competition.

*5.1. Tree-Based Ensemble Learning*

Decision trees [13] are nonparametric techniques used for classification and regression problems [14,15].

The outcome is modeled by a selection of independent covariates from a pool of variables. The feature space is recursively split to arrange samples with similar characteristics into homogeneous groups and maximize the difference among them. The tree is built starting from a root node, which is recursively split into daughter nodes according to a specific feature and split-point selection method. At each node of the tree, some statistical measure is minimized—commonly the (Gini) impurity [14], even though information gain [15] or variance reduction [14] can be used as well. The splitting finishes when a certain stopping criterion is met (e.g., early stopping [16]), or the full tree can be grown and pruned afterwards [17] to reduce overfitting bias.

However, a single tree is sensitive to small changes in the training data. This results into high variance in the predictions, making them not competitive in terms of accuracy [18,19].

Over the past decade, tree-based ensembles have become increasingly popular due to their effectiveness, state-of-the-art results, and applicability to many diverse fields, such as biology [20], chemistry [21,22], telecommunications, or games [1,2], among others. Tree-based ensemble methods [23] overcome the single-tree instability problem and also improve the accuracy of the results by averaging over the predictions from hundreds or even thousands of trees.

Decision tree ensembles are explicative models, useful to understand the predictive structure of the problem, and provide variable importance scores that can be used for variable selection [12].

In this work we focus in two tree-based ensemble approaches: extremely randomized trees and conditional inference survival ensembles.

5.1.1. Extremely Randomized Trees

Extremely randomized trees were first presented in [24] and consist of an ensemble of decision trees constructed by introducing random perturbations into the learning procedure from an initial subsample [25]. In this method, both the variable and the split-point selection are performed randomly based on a given cutting threshold. Totally randomized trees are built with structures independent of

the output values of the learning sample. This technique has been previously used in the context of video games [26].

Another popular randomization-based ensemble is the classical random forest algorithm [27], which differs from the extremely randomized forest in that the split point is not chosen randomly; instead, the best cutting threshold is selected for the feature. Because of this, the extremely randomized forest approach is faster, while producing similarly competitive results—even outperforming the random forest method in some occasions [24].

### 5.1.2. Conditional Inference Survival Ensembles

Conditional inference trees [28] are decision trees with recursive binary partitioning [29]. They differ from other variants in that they are formulated according to a two-step partitioning process: first, the covariate that is more correlated with the output is selected by standardized linear statistics; then, in a second step, the optimal split point is determined by comparing two-sample linear statistics under some splitting criterion. In this way, conditional inference ensembles [30] are not biased and do not overfit, contrary to one-step approaches for partitioning (e.g., random forests are biased towards variables with many possible split points [28]). Any splitting criterion can be used for the split, and the algorithm stops if a minimal significance is not reached when splitting [28].

Conditional inference survival ensembles [1,28] constitute a particular implementation of this method that uses the survival function of the Kaplan–Meier estimator as splitting criterion. Survival approaches were introduced to deal with time-to-event problems or censoring. Problems with censored data (of which churn is a typical example) are characterized by having only partially labeled data: we only have information about users who already churned, not about those who are still playing.

Other survival ensembles such as random survival forests [31] do deal with censoring, but they present the same kind of bias as the random forest approach. We refer the reader to [32] for a review on different types of survival trees.

### 5.2. Long Short-Term Memory

An LSTM network [33] is a type of RNN [34]. While conventional RNNs have a vanishing gradient problem that makes the network highly dependent on a limited number of the latest time steps [33] and are not able to remember events that occurred many time steps ago, LSTM networks consist of multiple gates trained to determine what information should be memorized or forgotten at each time step. As a consequence, an LSTM network can learn longer-range sequential dependencies than an RNN network.

LSTM networks have been successfully applied to e.g., machine translation [35], audio processing [34], and time-series prediction [36], achieving state-of-the-art results. They can be further extended to act like an autoencoder [9], learning without supervision a representation of time-series data to reduce the dimensionality or creating single vector representations of sequences [37]. Some other variations are e.g., bidirectional LSTM neworks that process both the beginning and end of the sequence [38] and gated recurrent units [39] that simplify the LSTM model and are generally more efficient to compute. There are other neural network approaches that have been explored in connection to video game data. For instance, deep belief networks were applied to time series forecasting in [40] and deep neural networks (DNNs) and CNNs served to predict customer lifetime value in [41].

## 6. Model Specification

For both tracks and test datasets, multiple models were analyzed, with those based on LSTM, extremely randomized trees, and conditional inference trees providing the best results—and leading us to win the two tracks. The parameters of each model were optimized by cross-validation. (See Section 7.1.)

*6.1. Binary Churn Prediction (Track 1)*

6.1.1. Extremely Randomized Trees

Extremely randomized trees proved the best model for the Test 1 dataset in this first track. A splitting criterion based on the Gini impurity was applied. After parameter optimization by cross-validation, 50 sample trees were selected and the minimum number of samples required to split an internal node was set to 50.

6.1.2. LSTM

In the first track, an LSTM model produced the best prediction results for the Test 2 dataset. To obtain representative features from the time-oriented data and static data mentioned in Section 3, a neural network structured with LSTM layers and fully-connected layers was proposed, that is, an LSTM network and a multilayer perceptron network (MLP) were combined. As shown in Figure 2, the time series input layer was followed by two LSTM layers, while static features were processed with a fully-connected layer. Then, the outputs from both parts were merged into a representative vector. Finally, an MLP network was applied to the representative vector to output the final result, the prediction on whether the user would churn or not.

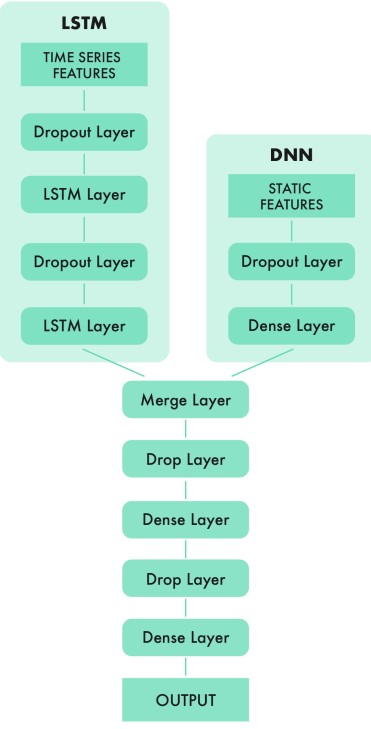

**Figure 2.** Neural network structure that provided the best results for the Test 2 dataset in the binary case (track 1). The left part is an LSTM network with the time-series data listed in Section 3.1 as input. The right part is a DNN that takes static features as input. The bottom part is a DNN which merges the outputs of these two networks and provides the final prediction result .

Dropout layers [42] were inserted in between every two layers to reduce overfitting and perform random feature selection. Since there were many correlated features, such as total playtime and daily playtime, randomly selecting a subset of the input time-series and static features by means of dropout layers prevented the model from depending too much on similar features and improved its generalizability.

*6.2. Survival Time Analysis (Track 2)*

Conditional Inference Survival Ensembles

Using conditional inference survival ensembles—a censoring approach—provided the best results for both test datasets in the second track. The following optimized parameters and settings were applied: 900 unbiased trees, 30 input features randomly selected from the total input sample, subsampling without replacement, and a stopping criterion based on univariate *p*-values.

The same approach was already used for video game churn prediction in [1,2]. However, in those works, the available data covered the whole player lifetime (since registration until their last login) and the maximum level reached and accumulated playtime up to the moment of leaving the game were also estimated. In contrast, in this competition we were working with partial user data, as previously explained in Section 2.2.

## 7. Model Validation

Model validation is a major requirement to obtain accurate and unbiased prediction results. It consists in training the model on subsamples of the data and assessing its ability to generalize to larger and independent datasets.

The data used in this study were affected by a business modification: a change in the billing system that shifted the distribution of the data, as explained in Section 4.2. Thus, as the training set consisted entirely of data taken from the subscription model (with free-to-play samples found only in one of the test sets), there was a significant risk of overfitting. For this reason, when performing our model selection, we either focused on methods that are not prone to overfitting (such as conditional inference survival ensembles) or tried to devise a way to avoid it (in the case of LSTM). For the extremely randomized trees approach, data preprocessing and careful feature selection are essential to reduce the prediction variance and bias by training the model on the most relevant features and discarding those that are not so important.

For model validation and estimation, we followed a *k*-fold cross-validation approach [43] with repetition.

Apart from the $F_1$ score and RMSLE mentioned in Section 2.3, additional error estimation methods were applied to evaluate models comprehensively.

*7.1. Cross-Validation*

In both tracks, we performed five-fold cross-validation for model selection and performance evaluation. The *k*-fold cross-validation [43] method reduces bias—as most of the data is used for the fitting—and also variance—since training data are used as the validation set. This method provides a conservative estimate for the generalizability of the model [44]. We decided to use 5 folds, as shown in Figure 3, so that each of them was large and representative enough, also making sure that they had similar distributions of the target data.

After partitioning the whole training sample into training and validation splits (consisting of 80% and 20% of the data, respectively), we performed the five-fold cross-validation for parameter tuning. With the best set of parameters, we applied the models to the validation split and selected only those that produced top scores.

Once the cross-validation process was finished, we trained the most promising models (using the adjusted parameters) on the whole training dataset and submitted our predictions for the Test 1 and 2 sets. After checking the partial results displayed in the leader board, we kept using only the models that were exhibiting the best performance. Tables 4 and 5 show our main results in the two tracks.

However, let us recall that the scores shown in the leader board were computed using only 10% of the total test sample. Thus, focusing too much in increasing those scores might have led to an overfitting problem, where the model would yield accurate predictions for that 10% subsample while

failing to generalize to the remaining 90% of the data. This is precisely what we wanted to avoid by performing an exhaustive cross-validation of the models prior to the submissions.

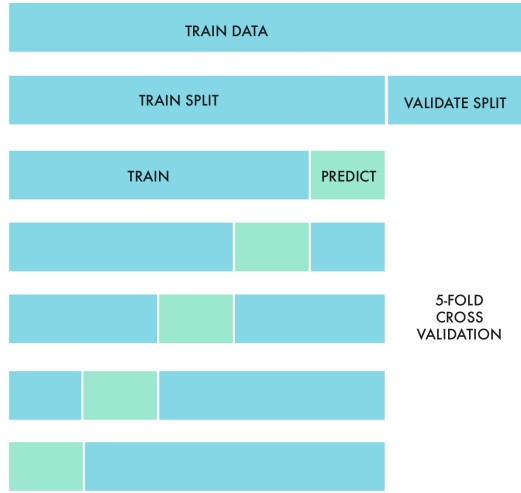

**Figure 3.** Five-fold cross-validation was performed after dividing the training data into training and validation splits.

**Table 4.** Track 1 results. (ERT: Extremely ranzomized trees, C. Ensembles: Conditional inference survival ensembles, LSTM: Combination of long short-term memory and multilayer perceptron networks, A: Feature value distribution, B: Feature importance, C: All features without feature selection, Validation: $F_1$-score on the training set, TP: True positive, TN: True negative, FP: False positive, FN: False negative, Accuracy: $(TP + TN)/(TP + TN + FP + FN)$.)

| Test Set | Model | Features | $F_1$ Score | Accuracy (%) | Validation | TP | TN | FP | FN |
|---|---|---|---|---|---|---|---|---|---|
| | ERT | A | 0.6096 | 73.23 | 0.650 | 627 | 1570 | 530 | 273 |
| | ERT | C | 0.6073 | 73.10 | 0.633 | 624 | 1569 | 531 | 276 |
| Test 1 | C. Ensembles | B | 0.6055 | 74.33 | 0.636 | 591 | 1639 | 461 | 309 |
| | LSTM | C | 0.5999 | 72.43 | 0.661 | 620 | 1553 | 547 | 280 |
| | LSTM | A | 0.5980 | 73.07 | 0.663 | 601 | 1591 | 509 | 299 |
| | LSTM | A | 0.6377 | 75.07 | 0.659 | 683 | 1654 | 446 | 330 |
| | C. Ensembles | B | 0.6173 | 75.00 | 0.638 | 605 | 1645 | 455 | 295 |
| Test 2 | LSTM | C | 0.6162 | 74.33 | 0.661 | 618 | 1612 | 488 | 282 |
| | ERT | A | 0.6159 | 75.13 | 0.639 | 598 | 1656 | 444 | 302 |
| | ERT | C | 0.6108 | 75.07 | 0.630 | 587 | 1665 | 435 | 313 |

**Table 5.** Track 2 results. (RMSLE: Root mean squared logarithmic error, RMSE: Root mean squared error, MAE: Mean absolute error, Validation: RMSLE on the training set.)

| Test Set | RMSLE | RMSE | MAE | Validation |
|---|---|---|---|---|
| Test 1 | 1.0478 | 127.0 | 104.9 | 0.9142 |
| Test 2 | 0.8621 | 60.9 | 45.3 | 0.9153 |

*7.2. Results*

Table 4 lists the track 1 results for the different models and feature selection methods (described in Section 4.3) that we explored. For the Test 1 dataset, the best results were achieved when using extremely randomized trees (ERT) and feature value distribution to perform feature selection. For the Test 2 dataset, the higher $F_1$-score was obtained with the model that combined LSTM and DNN networks and the same feature selection method as in the previous case.

In Table 5, we show the track 2 results. Here, we used a model based on conditional inference survival ensembles, and features were selected through the feature importance method.

It is interesting to note that, in both tracks, the Test 2 dataset yielded better results (higher $F_1$ scores in Table 4 and lower errors in Table 5) for all the models and feature selection methods tested—a trend also observed in the results obtained by other teams. Recalling that the Test 1 dataset covered a period in which the game followed a subscription-based business model, whereas for Test 2 the game was already free-to-play, this suggest that machine learning models learn player patterns more easily in games that employ a freemium scheme.

From these observations, it is apparent that the ability of the models to generalize and adapt to evolving data, namely to learn from different data distributions, was essential in this competition. Due to this, we devoted much effort to perform a careful model selection and validation, which allowed us to clearly outperform all the other groups at the end—when the full test sample was used to obtain the final scores for each model—even if we were not at the top of the leader board.

### 7.3. Binary Churn Prediction (Track 1)

Besides the $F_1$ score, we also used the receiver operating characteristic (ROC) curve and the AUC measure to provide a more exhaustive validation.

The ROC curve is constructed by representing the true positive (accurately predicted positive samples) and false positive (misclassified negative samples) rates, and the AUC is just the area under the ROC curve [45].

### 7.4. Survival Time Analysis (Track 2)

In the case of track 2, the outcome is the overall survival time of the player in the game. In addition to the RMSLE measure, we evaluated the mean absolute error (MAE), root mean squared error (RMSE) and integrated Brier score (IBS). The MAE describes the average deviation of the predicted value with respect to the actual value. The RMSE has the same expression as the RMSLE (see Equation (4)) but without taking logarithms. Finally, the IBS is an evaluation measure specifically formulated for survival analysis [46,47] that provides a final score for the error estimation of the survival time outputs.

## 8. Discussion and Conclusions

The models that led us to win both tracks of the Game Data Mining competition hosted at the 2017 IEEE Conference on Computational Intelligence and Games were based on long short-term memory networks, extremely randomized trees, and conditional inference survival ensembles. Model selection was certainly key to our success in track 2, which involved predicting the time for each player to churn (i.e., leave the game), as we were the only group that employed a model well-suited to deal with censored data. Since censoring is inherent in churn, adopting a survival model gave us a decisive advantage.

All the selected models proved to be robust against evolving conditions, managing to adapt to the change in the game's business model that took place between the two provided datasets. In addition, they were efficient in terms of time cost: in both tracks, it took less than one hour to train each model on the 4000 user training set and apply it to the 6000 testing users. (As cross-validation and intensive parameter optimization were performed, the full experimental procedure actually took longer.) Thanks to these properties—and also to their ability to scale to large datasets—our models could be readily implemented in real business settings.

There were some factors that made this competition particularly challenging. First, only 10% of the test data were available for validation, which forced us to perform an exhaustive and careful training in order to prevent overfitting. Moreover, the fact that the provided datasets contained information on just a small subsample of all players of this title and covered a short period of time (6–8 weeks) made this prediction problem particularly ambitious. Knowing the characteristics of the selected players (e.g., whether all of them were paying users or even VIP users) or the full data history

of each player—since they registered until their last login—could have served to achieve even better results, as those obtained in [1,2] for a model based on conditional inference survival ensembles.

Computing additional features could also have helped to improve accuracy. Concerning the machine learning models, it would be interesting to apply other advanced DNN techniques, such as CNNs, which were already used in [41] to predict the lifetime value of video game players from time series data.

On the whole, by winning this competition we showed that the proposed models are able to accurately predict churn in a real game, pinpointing which players will leave the game and estimating when they will do it. Moreover, the accuracy of the results should be even higher in real situations in which some of the limitations discussed above are removed—in particular, once larger datasets that include more users and span longer periods of time are available. Thanks to this accuracy and also to their adaptability to evolving conditions, time-efficiency, and scalability, we expect these machine learning models to play an important role in the video game industry in the forthcoming future, allowing studios to increase monetization and to provide a personalized game experience.

**Author Contributions:** Conceptualization, A.G., P.P.C. and Á.P.; Methodology, A.G., P.P.C. and Á.P.; Software, A.G., P.P.C. and Á.P.; Validation, A.G. and P.P.C.; Formal Analysis, A.G., P.P.C. and Á.P.; Investigation, A.G., P.P.C. and Á.P.; Resources, Á.P.; Data Curation, P.P.C.; Writing—Original Draft Preparation, A.G., P.P.C. and Á.P.; Writing—Review & Editing, A.G. and Á.P.; Visualization, A.G. and P.P.C.; Supervision, Á.P.; Project Administration, Á.P.

**Funding:** This research received no external funding.

**Acknowledgments:** We thank Javier Grande for his careful review of the manuscript and Vitor Santos for his support.

**Conflicts of Interest:** The authors declare no conflict of interest.

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
