# Peer review of "The Winning Solution to the IEEE CIG 2017 Game Data Mining Competition"

_make, doi:10.3390/make1010016_

Round 1
Reviewer 1 Report
First of all, I'd like to congratulate authors for winning the competition.
The paper is interesting, but need to be improved.
Both in the introduction there is too much information on the context and very little in the solution and results obtained.
Using thousands separator in talbe 1 and the text would improve reading experience
The "G" in size column (Tablet 1) in unclear, use GB or GiB instead
The fact of the 6 or 8 weeks is unclear (was there any pattern in the missing data or any reason fir it?). And, more interesting, its possible effect in the algorithm is not commented.
In section 2.3 authors comment how track 1 was evaluated. What about track 2?
Information in section 3 has too many subsection with very little information, consider using a better explanation, using bullets or a table
Using a reference as a name (section 2.3: "[14] used a deep belief network") reads very poorly, improve the phrase.
In section 2.3: is there any approach of using those techniques in a problem similar to the one in this paper. If so, comment. If not, explain that is a novel application.
Between section 5 and section 5.1 a short phrase would improve reading: "In this section we introduce the background ..."
Consider if Figure 2 would improve adding a square for left part (LSTM and other for right one (DNN)
Table 3 is too large, using an acronym for "Conditional Inference Ensembles" would solve problems.
Given that the reference for the value of the approach is winning the competition, authors should dedicate a section (or subsection) to explain results, their difference to others in the ranking and other information about the competition.
There is a need for a "Threats to validity", commenting limitations in the approach. Additionally, specially for a special issue like this a comment on the gender of players (better if actual data is provided, if not, refering a study on it) could affect the validation of the results.
Additionally, there is no clear conclusion. There is a good work, but conclusions have to be summed up at the end of the text.
In the bibliography the providing the arxiv prerint is great, but the reference need for citation (book conference proceedings, paper, etc).
Additional comments:
* Missing affiliation of third author
* Missing keywords
* Bibliography: review carefully: CIG => "Computational Intelligence and Games", ieee => IEEE, lstm => LSTM, etc
Author Response
Response to Reviewer 1 Comments
First of all, I'd like to congratulate authors for winning the competition.
The paper is interesting, but need to be improved.
Both in the introduction there is too much information on the context and very little in the solution and results obtained.
Thank you very much for your congratulations and your feedback. While it is true that we devote most of the introduction to the context, we think this background is important to understand the problem addressed in the competition and its importance, as well as to introduce the machine
learning methods we used to win it. At the same time, we believe it is clear both from the Abstract and the Introduction that our winning solutions were based on a long short-term memory network and a conditional inference survival ensemble model and that it is reasonable to defer the specifics of these models to later sections.
Using thousands separator in table 1 and the text would improve reading experience
The "G" in size column (Tablet 1) in unclear, use GB or GiB instead
Thank you for your pertinent remark, which will surely improve the readability of the data. Both changes have been introduced in Table 1.
The fact of the 6 or 8 weeks is unclear (was there any pattern in the missing data or any reason for it?). And, more interesting, its possible effect in the algorithm is not commented.
Thank you for your remark. We understand that here you are referring to Figure 1 and its caption. As indicated in Table 1, the training dataset covered a period of 6 weeks, whereas each of the 2 test datasets spanned 8 weeks. This what we mean by that “6 or 8 weeks” that can be seen in Figure 1. Therefore, it is not that we chose whether to use 6 or 8 weeks, but rather just the characteristics of the provided datasets. To make this clearer, we have added a reference to Table 1 in the caption of Figure 1.
In section 2.3 authors comment how track 1 was evaluated. What about track 2?
Thank you for your question. The explanation on how track 2 was evaluated is also contained in section 2.3 (see Equation (4) and the paragraph it belongs to).
Information in section 3 has too many subsection with very little information, consider using a better explanation, using bullets or a table
Thanks a lot for bringing this to our attention. Apparently, during the submission procedure the format of some tables and sections in the document was automatically changed, and Section 3 was one of the affected parts. We have amended the format of Section 3.1 (pages 3-4) so that
now it is displayed as a list.
Using a reference as a name (section 2.3: "[14] used a deep belief network") reads very poorly, improve the phrase.
Thank you for your pertinent remark. You are right, indeed, and this has been amended in page 4, line 99 (Section 3.2).
In section 2.3: is there any approach of using those techniques in a problem similar to the one in this paper. If so, comment. If not, explain that is a novel application.
Thank you for your relevant comment.This is not a novel application (please note that, as this is a competition paper, its main purpose is to show the state of the art in a particular kind of problem and the methods, algorithms or applications do not have to be necessarily novel). You are right in that it is indeed appropriate to comment on the use of these techniques in similar
problems, and we did so by introducing some sentences in Section 5, when explaining the different models. See page 5, lines 166-167 and page 6, lines 203-205.
Between section 5 and section 5.1 a short phrase would improve reading: "In this section we introduce the background …"
Thank you for your pertinent remark. We have included such a short sentence in page 5, line 138 (Section 5).
Consider if Figure 2 would improve adding a square for left part (LSTM and other for right one (DNN) problems.
Thank you for your appropriate remark. We modified Figure 2 and we hope that is clearer now.
Table 3 is too large, using an acronym for "Conditional Inference Ensembles" would solve problems.
Thank you for bringing this to our attention. As previously commented, during the submission procedure the format of some tables and sections in the document was automatically changed, and Table 3 was affected. We have followed your suggestion and substituted “C. Ensembles” for “Conditional Inference Ensembles”, as can be seen in page 9, Table 5.
Given that the reference for the value of the approach is winning the competition, authorsshould dedicate a section (or subsection) to explain results, their difference to others in the ranking and other information about the competition.
Thank you for your relevant comment. Following your suggestion, we have introduced a “Results” subsection in page 9, line 278 (Section 7.2).
There is a need for a "Threats to validity", commenting limitations in the approach. Additionally, specially for a special issue like this a comment on the gender of players (better if actual data is provided, if not, referring a study on it) could affect the validation of the results.
Thank you for your feedback. We have introduced a paragraph commenting on some limitations related to the special characteristics of the competition in the “Discussion and Conclusions” section, in page 10 of the manuscript, lines 325-332. On the other hand, gender of players was not specified in the original dataset and thus was not a relevant factor for the validation or predictions of our models.
Additionally, there is no clear conclusion. There is a good work, but conclusions have to be summed up at the end of the text. → Add conclusions (also that we were the only group considering the censoring case and selecting a model that handles it).
Thank you for your relevant comment, which surely served to improve the presentation of the manuscript. This has been amended in the Discussion and Conclusions section, on page 10. In particular, following your suggestion, we have stressed that we are the only group that considered censoring and used a model able deal with it (see page 10, lines 313-317).
In the bibliography the providing the arxiv preprint is great, but the reference need for citation (book conference proceedings, paper, etc).
Thank you for bringing this to our attention. All the affected references have been amended, please see line 370 on page 11 and lines 395, 408, 411, 429 on page 11.
Additional comments:
* Missing affiliation of third author
This has been amended.
* Missing keywords
This has been amended.
* Bibliography : review carefully: CIG => "Computational Intelligence and Games", ieee => IEEE, lstm => LSTM, etc
All have been amended, in page 11 line 378 and in page 12 lines 384, 387, 429.

Reviewer 2 Report
The authors should describe with pseudocode the winning algorithms.
The authors should present some information about the time efficiency of the examined methods.
Author Response
Response to Reviewer 2 Comments
The authors should describe with pseudocode the winning algorithms.
Thank you for your feedback. Whereas it would be undoubtedly very enriching for the paper to include the pseudocode of the winning algorithms, as we work in industry, we cannot disclose this information due to an NDA.
The authors should present some information about the time efficiency of the examined methods.
Thank you for your remark. Indeed, discussing the time efficiency of the methods is highly relevant, and so we have included a comment about it in the Discussion and Conclusions section (see page 10, lines 318-344).

Round 2
Reviewer 1 Report
Congratulations to the authors for the improvement of the paper.
My only recommendation is that they add more information about the conclusions of the paper in the abstract, to make it self-contained. In the new version of the paper, authors improved the conclusions, some of that information could improve the last part of the abstract.
Author Response
Response to Reviewer 1 Comments
Congratulations to the authors for the improvement of the paper.
Thank you very much.
My only recommendation is that they add more information about the conclusions of the paper in the abstract, to make it self-contained. In the new version of the paper, authors improved the conclusions, some of that information could improve the last part of the abstract.
Thank you for your valuable remark. We have extended the Abstract including some of the conclusions.